



# Assessing ecosystem services under water stress in the largest inland river basin in China based on hydro-ecological modeling

Yang Yu,[1,2] Markus Disse,[3] Philipp Huttner,[3] Xi Chen,[4] Andreas Brieden,[5] Marie Hinnenthal,[5] Haiyan Zhang,[1] Jiaqiang Lei,[1] Fanjiang Zeng,[2] Lingxiao Sun,[6] Yuting Gao,[6] Ruide Yu,[1,7]*

[1]State Key Laboratory of Desert and Oasis Ecology, Xinjiang Institute of Ecology and Geography, Chinese Academy of Sciences, Urumqi 830011, China;
[2]Cele National Station of Observation and Research for Desert-Grassland Ecosystem, Cele 848300, China;
[3]Chair of Hydrology and River Basin Management, Technical University of Munich, Munich 80333, Germany;
[4]Research Center for Ecology and Environment of Central Asia, Chinese Academy of Sciences, Urumqi 830011, Xinjiang,
China;
[5]Chair of Statistics and Risk Management, Universitaet der Bundeswehr Muenchen, Neubiberg D-85577, Germany;
[6]University of Chinese Academy of Sciences, Beijing 100049, China;
[7]School of Environment and Material Science, Yantai University, Yan Tai 264005, China

*Correspondence to*: Ruide Yu (ruideyu@ms.xjb.ac.cn)

**Abstract.** Ecological changes in arid lands are often determined by the available water resources. For ecological protection, it is important to consider both hydrological and ecological processes. This paper presents a modeling approach to assess environmental changes and ecosystem services (ESS) in the largest inland river basin in China. Current water cycle and ecosystem protection measures were simulated, and future land use change scenarios were proposed accordingly. China is on the frontline of ecosystem protection and afforestation, but according to the simulation results, the available water resources

cannot support more vegetation in its largest inland river basin. Without an additional water supply, 25.9% of the existing area of natural vegetation will be degraded by 2050. A reduction in the area of cotton plantation did not substantially reduce farming incomes in the simulation. Drip irrigation and ecological flooding was effective for improving ESS, especially agricultural provisioning and riparian forest provisioning services, as well as regulating and supporting services. The assessment of ESS can achieve socio-economic and ecological benefits in a sustainable way, and also enable the impact of

the complex environmental factors to be understood. The results can be used to evaluate current situations and develop creative solutions.

## 1 Introduction

Ecosystems provide a range of goods and services for human society, which are collectively known as ecosystem services (ESS). Land use changes can drive the transformation of ESS, and in arid lands they are further determined by human

activities and water availability. On a regional scale, there is an increasing consensus about the importance of incorporating ESS into basin management decisions (Sawut et al. 2013; Hu et al. 2015; Zhang et al. 2016). However, it has proven difficult to quantify the levels and values of ESS (Nelson et al. 2009), and it is even more difficult to predict and assess ESS changes



because it requires the simulation of both hydrological and ecological processes. To better understand the physical processes of ecological changes, it is necessary to establish models that focus on available water resources in arid lands (Schwinning and Ehleringer 2001; Kahil et al. 2015). Due to model complexity and computational time, it is very difficult for a single model to consider both hydrological processes and ESS changes (Mcdonnell et al. 2007; Warton et al. 2015). In most cases, the water cycle only needs to be simulated once (after calibration and validation), but the simulation of ecosystem changes and management alternatives need to be repeated many times until several optimized sustainable management solutions can be identified. Therefore, multiple models running in parallel is desirable for simulating regional hydrological and ecological dynamics in arid lands.

China has made a great effort to ensure ecosystem protection in recent years (Xu et al. 2006; Liu et al. 2010; Cao et al. 2015; Li 2017). Some definite improvements have been made regarding the ecological problems in many regions (Bai et al. 2006; Zhang and Liu 2009; Gao et al. 2010; Yin 2014). However, some environmental problems still persist even after the implementation of ecological rehabilitation projects or after several years of remedial action (Wan 2003; Hua et al. 2006; Feng et al. 2015; Zhao et al. 2017), especially in arid lands. The assertion that, "Lucid waters and lush mountains are invaluable assets", has been reported and emphasized officially many times since the 19[th] National Congress of the Communist Party of China (Fu et al. 2019). Due to government determination and the aspiration of the people, a new era of ecosystem protection has been predicted to emerge in China (Lei et al. 2018). Nevertheless, a current report from Nature warned that China's tree-planting could strain water resources (Mark, 2019). Water availability sets an implicit limit on unfettered ecological expansion in arid regions. Surface and groundwater shortages threaten ecosystem protection and ecological restoration in the arid and semi-arid lands of China (Lu et al. 2016). Arid ecosystems are characterized by great variability and vulnerability (Lioubimtseva and Henebry 2009; Liu et al. 2016). Effective arid ecosystem management requires a good understanding of how the system works (Hart et al. 2010), and how it can be affected by environmental changes and human activities under water scarcity.

The Tarim River Basin is the largest inland basin in China, with a total area of $1.05 \times 10^6$ km$^2$, which is about 10.9% of the land area of China. Located in a dryland area of central Asia, the desert area in the basin is $3.7 \times 10^5$ km$^2$. The Tarim River Basin has been extensively investigated by the scientific community due its unique environment (Zhao et al. 2006; Huang et al. 2010; Cyffka et al. 2013). Annual precipitation in the basin is around 50 mm, while potential evaporation reaches 3000 mm each year. In such arid areas, water is crucial for environmental changes and ecosystem protection. Land use changes are often closely intertwined with changes in water resources (both surface water and groundwater). Therefore, the simulation of the water cycle has enabled a better understanding of land use changes in recent years, as well the planning of ecosystem protection in the future. The Tarim River is a typical seasonal river in an arid area, with floods in the summer and dry seasons in spring, autumn, and winter. The average annual discharge in the mainstream during the last 60 years was approximately $4.5 \times 10^9$ m$^3$ (Yu et al. 2015). Due to overexploitation and other anthropogenic activities, water interception and severe environmental problems have occurred in the river oases (Hou et al. 2009; Wang et al. 2013). Excessive use of water in the upper reaches been ongoing for several decades, resulting in the reduction of available water resources



downstream. High evapotranspiration rates and high groundwater salinity levels can intensify water shortages (Pang et al. 2010). In 2018, the total afforested area of the Tarim River Basin was 5,200 km$^2$. It is not clear if the available water resources are able to support the establishment of new vegetation in the river oases.

The Tarim River Basin has developed as a vast, unique, and variable natural landscape during its long history (Wu and Cai 2004; Zhao et al. 2013). The world's largest natural *Populus euphratica* specimens are distributed in the basin oases (Thomas et al. 2017). The Taklimakan Desert is the second largest desert in the world, and the mysterious Lop Nor has long been attractive to adventure tourists (Tao et al. 2016). Due to the hyper-arid climate, the natural vegetation (other than *P. euphratica*) is mainly comprised of low bushes, including *Achnatherum splendens*, *Apocynum* spp., *Tamarix chinensis*,

*Halimodendron halodendron*, *Kalidium foliatum*, and *Nitraria tangutorum*. These drought-tolerant plants can alleviate desertification conditions to a certain extent (Yu et al. 2015). In the region's arable land, the soil types are mainly loamy sand. Because of the high evapotranspiration, irrigation water shortages, and poor drainage systems, soil salinization has been aggravated in recent years. Statistical data has revealed that 38% of the region's agricultural fields suffer from salinization (Xu et al. 2008). The most serious salt contaminated area is in the middle and lower parts of the Tarim River (Xu

et al. 2014). Economic development in the Tarim River Basin is far behind that of eastern China. The basin is dominated by agriculture and grazing. In recent years, petroleum exploration and the petrochemical industry have been developed (Huang et al. 2016), but most of the local populace have not been benefited from this activity. In 2000, the Chinese central government initiated the western development strategy (Lu et al. 2013), and a huge amount of investment was made to develop the western regions of China, including the Tarim River Basin (Li et al. 2011). New railways are currently under

construction and the basin economy is developing rapidly (Tong et al. 2015). Under such circumstances, the utilization of water resources and protection of the ecological environment are more important than ever before. Population growth and farmland reclamation have substantially changed the land use, water resources, and ecosystems in the basin (Zhao et al. 2013). In the mainstream of the Tarim River, the landscapes are very different from those of the upper to lower reaches. Most population, farmland, and industries in the river oases are located in the upper reaches (Feike et al. 2015). Riparian

forest and scattered desert plants are mainly distributed in the middle and lower reaches. How best to evaluate ESS changes in this arid basin remains uncertain, but will have a large influence on decision-making and the sustainable development of the whole region. This study adopted a hydro-ecological modeling approach to assess ESS changes, while considering water availability. The results provide further guidance for regional sustainable development and ecosystem protection in arid lands.

## 95   2 Method

To simulate hydro-ecological processes and assess ESS changes, two hydrological models (MIKE HYDRO and MODFLOW) were employed and a decision support system (DSS) was established based on Qt Creator and C++ programming. Meteorological, hydrological, geographical, ecological, and socio-economic data were collected as inputs into





the models, and the ESS evaluations were provided by DSS outputs. The connection between MIKE HYDRO and the DSS
was at the sub-catchment level with regard to the surface water consumption and allocation strategies. Groundwater
conditions were simulated by MODFLOW and were fed to the DSS on a monthly basis. The equations and other logical
inputs in the DSS were created by reference to expert knowledge in cross-disciplinary research fields. The massive amounts
of research data used in the study were based on years of sample collections from field stations and observations, supported
by the Sino-German Sustainable Management of River Oases Along the Tarim River (SuMaRiO) project, which is a
collaboration between 11 German and 9 Chinese research institutions.

Ecosystem services are usually comprised of provisioning, regulating, and supporting services. In this study, ESS were
divided into three classes: services from agriculture, riparian forest, and grassland (Fig. 1). Agriculture can provide
provisioning services, including cotton production, fruit production, production of other crops, and farming incomes.
Riparian forest provides provisioning services (biomass production), regulating services (drifting dust control, sand
stabilization, wind speed reduction, and carbon sequestration), and supporting services (tree species). Grassland provides
provisioning services (*Apocynum* and reed production) and regulating services (drifting dust control and sand stabilization).
In the first edition prototype of the model, only the major functions of the ESS were considered. This reduced the
computational time required for each scenario. If enough knowledge and data are collected in the future, it would be possible
to add other ESS functions, such as regulating the regional climate, protection of wild animals, and flood control.

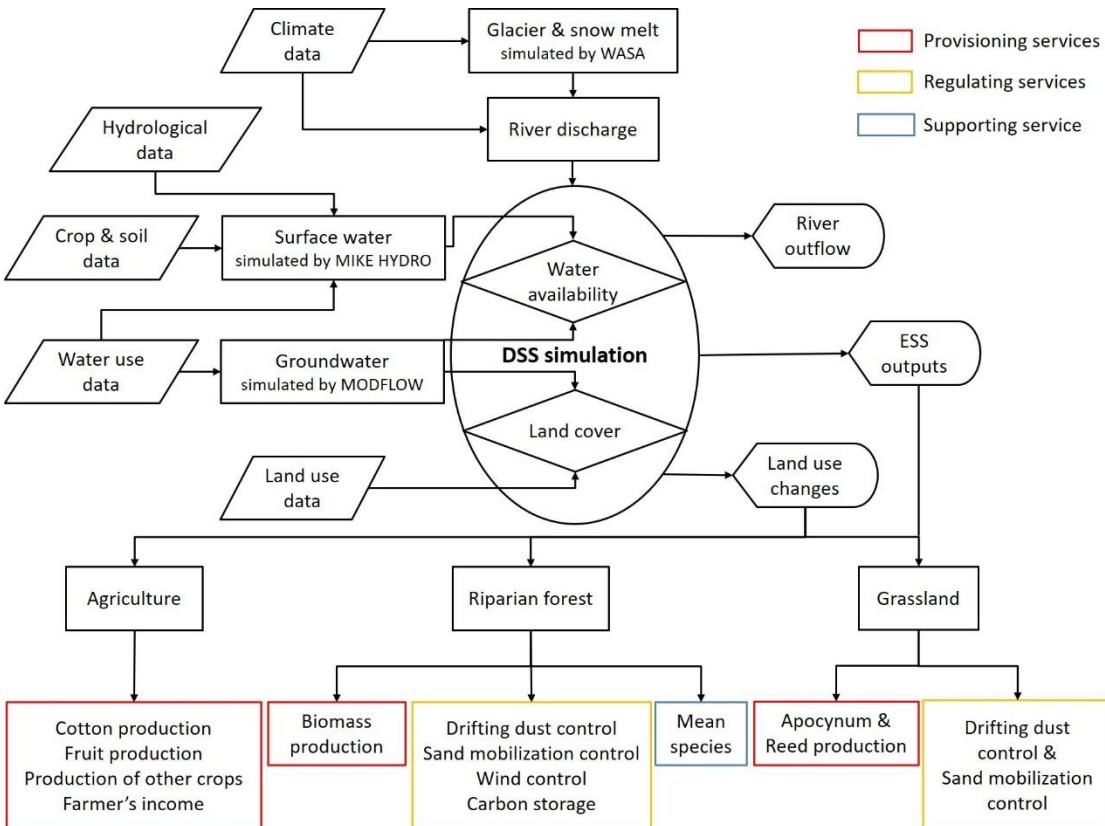




**Figure 1: Flow chart of the hydrological and ecological simulation process. Ecosystem services were divided into three categories of provisioning, regulating, and supporting services.**

MIKE HYDRO model is a versatile and physically-based modeling tool for the analysis, planning, and management of integrated water resources within a river basin (DHI 2014). The model used in the Tarim River Basin (Fig. 2) contained four sub-catchments (namely SC1, SC2, SC3, and SC4), a mainstream with Muskingum routing (Chow et al. 1988), a simplified one-layer groundwater module, a Food and Agriculture Organization (FAO) dual crop coefficient module, an FAO 56 climate module using the Penman-Monteith method for evapotranspiration (Allen et al. 1998), an FAO 56 soil module with nine different soil types, more than 50 water users (for irrigation, ecosystem, livestock, domestic, and industrial uses) in the oases, and eight reservoirs to regulate seasonal surface runoff. The MIKE HYDRO model was fully calibrated and validated over a nine-year period using observed discharges from gauging stations, and a high level of agreement was attained between simulated and observed values (Yu et al. 2017). The MIKE HYDRO model provided information for irrigation water demand, ecological water, infiltration, and fruit and crop production in the DSS simulation. The MODFLOW model was applied in parallel to simulate groundwater conditions over the same period. MODFLOW is a commonly used three-dimensional modular groundwater model (Harbaugh et al. 2000). The MODFLOW model in the Tarim River Basin provides an updated water head, river leakage, irrigation seepage, ecological water percolation, and other simulated groundwater movements using $500 \times 500$ m cells.

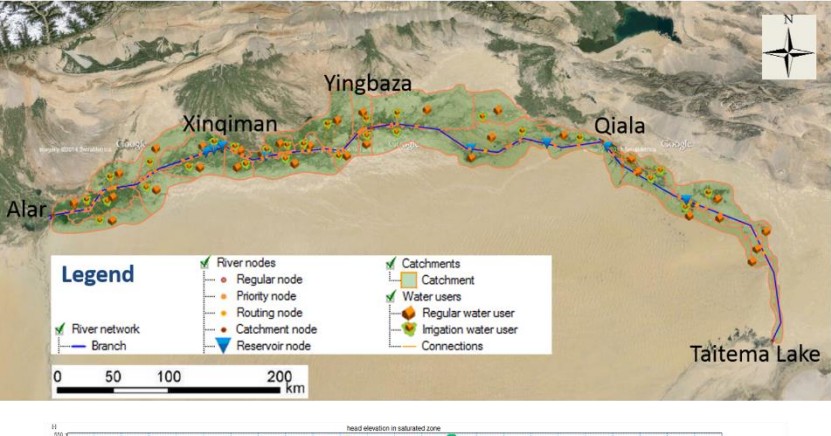

(a) Surface water distribution model: MIKE HYDRO model map view (Yu et al., 2017). The irrigation water demand, ecological water for the natural vegetation, seepage losses, fruit production, and crop yields were calculated and summarized as inputs into the DSS.

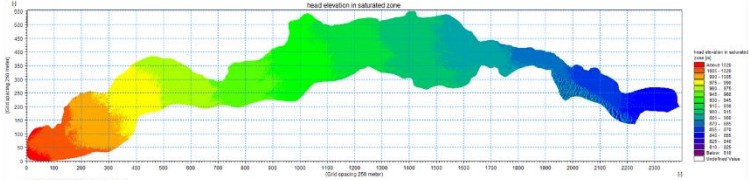

(b) Groundwater model: MODFLOW model initial water head. Cell size 500 by 500 m. Simulation running without in the lower reaches where no farmland area or riparian forest exist.

**Figure 2: Hydrological models used in the study: a) MIKE HYDRO model for surface water distribution; b) MODFLOW model for groundwater simulation.**

Both the MIKE HYDRO and MODFLOW models were fully calibrated and validated to precisely simulate the water cycle. Surface flow was calibrated from 2005 to 2009 with daily hydrological station data, and validated from 2010 to 2013 with weekly observed values. The groundwater level was calibrated and validated with monthly values from observation





wells. To build connections among hydrological processes, ecological processes, and ESS outputs, a certain logic must be established in the DSS. The logical inputs mainly include mathematical equations, computer algorithms, and fuzzy logics

(Mendel, 1995). These logical inputs were formed by geoscience theorems, empirical equations, experimental outcomes, and expert knowledges. Many empirical equation need parameter modifications before they can be applied in this arid region (Patrick et al. 2015; Yu et al. 2017). In addition to the scientific experts, many local farmers and stakeholders were also invited to provide their opinions on the socio-economic developments in the basin. Their advice and suggestions were collected and considered in the management scenarios and future ESS changes. In total, there were more than 100 equations,

200 data files, and 10,000 lines of C++ program in the DSS.

The river discharge was based on observation data and was simulated by the Water Availability in Semi-Arid Environments (WASA) model in the upstream mountainous region for future projections. River outflow was determined by inflow and water consumption along the oases (with no rainfall-runoff in the mainstream Tarim River). Farming incomes were calculated by cotton, fruit, and other crop production multiplied by the profit of each crop's production. Cotton

production was aggregated from the cotton yields in each cell, and further determined in terms of water availability, soil salinity, and an evapotranspiration–yield relationship model (Stewart et al. 1977; Yu et al. 2015). Drifting dust control, sand stabilization, and wind speed reduction were all determined in terms of tree height and crown area by fuzzy rules in each cell, and then aggregated to calculate a total value. Tree species were determined by the fuzzy logic between groundwater level and the flooding of natural vegetation. *Apocynum* and reed production were influenced by groundwater level, groundwater

salinity, and grazing area. Drifting dust control and sand stabilization by grassland were determined by grassland area and density. The simulation period was from 2012 to 2050, with monthly time steps.

Water-saving irrigation is a watering strategy that can be applied using different types of irrigation application methods. The correct application of water-saving irrigation requires a thorough understanding of the yield response to water (crop sensitivity to drought stress) and the economic impact of a reduction in the harvest (English 1990). In regions where water

resources are restricted it can be more profitable for a farmer to maximize crop water productivity rather than maximizing the harvest per unit land (Fereres and Sorian 2007). In the study area, the implementation of drip irrigation under mulch is becoming popular to save water and guarantee productivity. Our model (Fig. 3) simulated the application of drip irrigation by considering the wetting fraction ($f_w$) on the soil surface and infiltration depth of the wetted parts ($I_w$). Drip irrigation under mulch is mainly applied in cotton fields, with the ground surface being approximately 80% covered by transparent polythene

mulch (Zia-Khan et al. 2014). The wetting fraction was reduced from 1 in sprinkler irrigation to 0.1 in our drip irrigation module. This substantially reduced the spray losses, and the saved water could be used for ecological flooding or other purposes.

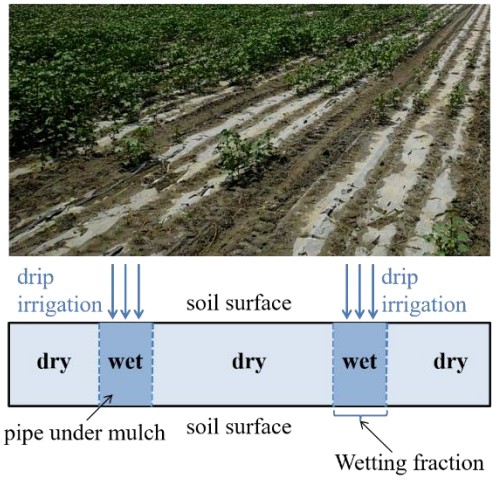

drip
irrigation | soil surface | drip
irrigation

| dry | wet | dry | wet | dry |

pipe under mulch   soil surface

Wetting fraction

(a) Horizontal illustration of drip irrigation under mulch in the research area.

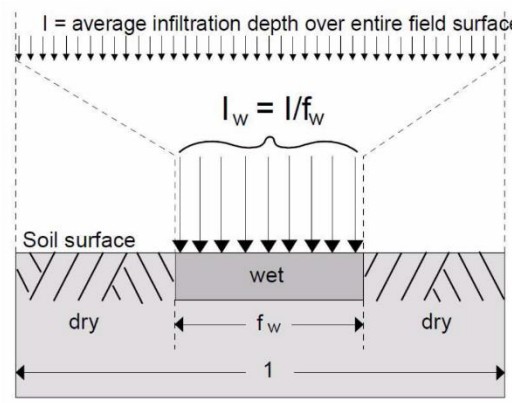

(b) Vertical illustration of drip irrigation, where I and I$_w$ are the irrigation depth for the entire field and the irrigation depth for the part of the wetted surface, respectively, and f$_w$ is wetting fraction (DHI, 2014).

**Figure 3: Drip irrigation under mulch: a) horizontal illustration of the research area; b) vertical illustration of the model.**

Additionally, a wide range of anthropologic activities can impact upon ESS. In our model, these activities were summarized to provide a series of major management alternatives (Fig. 4): land use changes (both actual and simulated), water consumption by households and industry, flooding of natural vegetation, the amount of drip irrigation applied on irrigated farms, and subsidies for the farmers in the four sub-catchments. Users or decision-makers can change the reference values of the management alternatives in a planning year. Any combination of input values defined by the users is possible

for the management scenarios. There were seven land use types depicted in the editable map (cells), with a reference land use map generated from MODIS data in 2012.

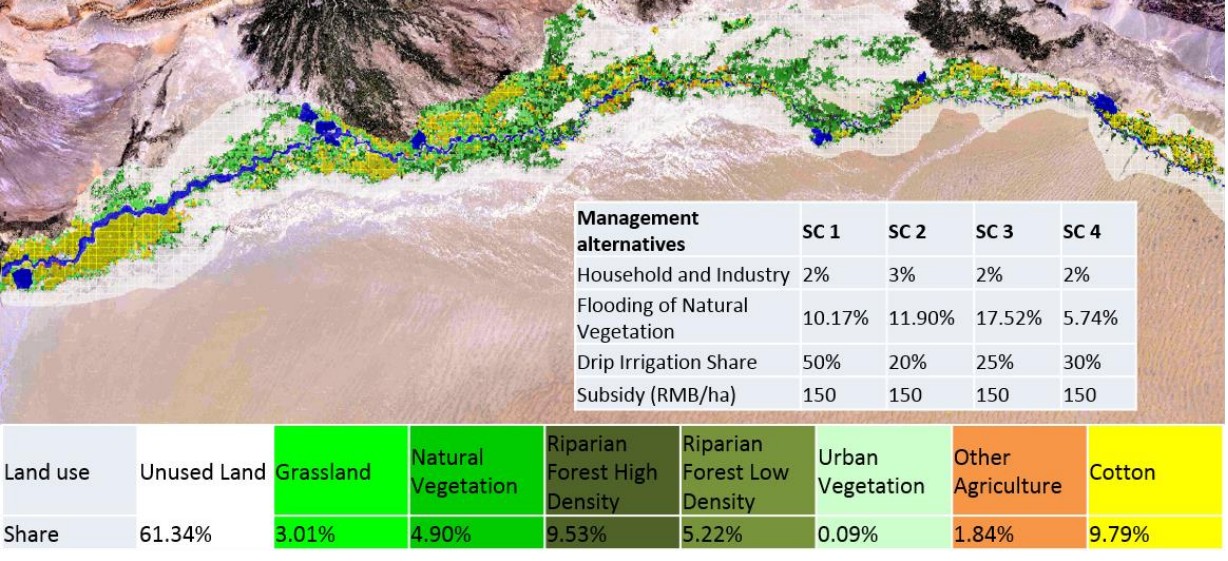

| Management alternatives | SC 1 | SC 2 | SC 3 | SC 4 |
|---|---|---|---|---|
| Household and Industry | 2% | 3% | 2% | 2% |
| Flooding of Natural Vegetation | 10.17% | 11.90% | 17.52% | 5.74% |
| Drip Irrigation Share | 50% | 20% | 25% | 30% |
| Subsidy (RMB/ha) | 150 | 150 | 150 | 150 |

| Land use | Unused Land | Grassland | Natural Vegetation | Riparian Forest High Density | Riparian Forest Low Density | Urban Vegetation | Other Agriculture | Cotton |
|---|---|---|---|---|---|---|---|---|
| Share | 61.34% | 3.01% | 4.90% | 9.53% | 5.22% | 0.09% | 1.84% | 9.79% |





**Figure 4: Management alternatives in the decision support system (DSS). The editable land use map enabled land use types to be changed for different planning years.**

The development of the DSS involved contributions from a variety of disciplines, such as computer science, hydrology, geography, pedology, botany, agronomy, humanities, and economics. By including this cross-disciplinary knowledge in the DSS, the links between the environmental elements were reproduced and the resulting scenarios enabled future predictions to be made.

## 3 Results

### 3.1 Water availability

The outputs of the hydrological models (Fig. 5) indicated that water availability was very limited in spring, autumn, and winter, but there was a substantial improvement in the summer. For surface water, summer floods (in July, August, and September) accounted for 78.8% and 75.2% of the total river discharge in 2012 and 2013, respectively. There was little surface runoff in winter and spring, especially in SC2, SC3, and SC4. There was an obvious reduction in the discharge from
upstream to downstream. The discharges in SC4 were only 13.7% and 8.9% of those in SC1 in 2012 and 2013, respectively. Huge amounts of water resources were consumed by agriculture and natural vegetation in the upper and middle reaches. In addition, because the water volume in 2013 was lower than in 2012, a higher percentage of the surface water would have been used in the upper and middle reaches than in a wetter year, which would in turn intensify downstream water shortages in the dry year. On the other hand, groundwater conditions were also better in summer than in winter. The water heads were
raised by 2.51 and 2.74 m from February to August in 2012 and 2013, respectively. As with the surface water, the groundwater availability was also much poorer in the middle and lower reaches than it was in the upper reaches, especially in winter. Groundwater levels were lower than 10 m in a parts of SC4. This result concurs with some of the previous findings reported in this region (Feng et al. 2005; Chen et al. 2015), and indicates that in some regions of the lower reaches, the groundwater table is beyond the reach of most desert vegetation (Fan et al. 2004; Hao et al. 2010).

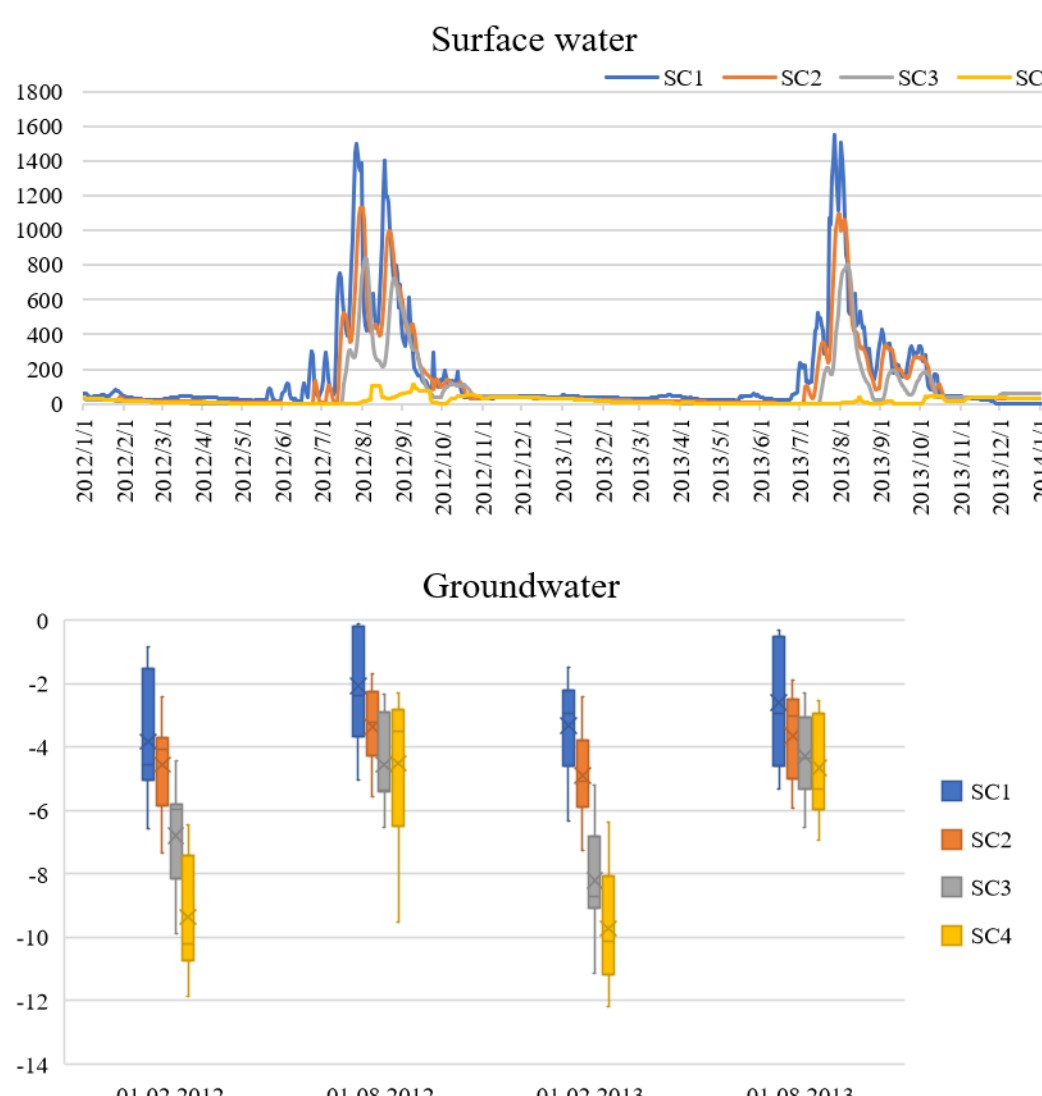


**Figure 5: Water availability in 2012 and 2013. Both models were fully calibrated and validated using observation data from gauging stations and wells.**

In SC4, groundwater levels were raised during the summers of 2012 and 2013. This was caused by the implementation of a water conveyance project to the lower reaches (Xu et al. 2008; Chen et al. 2010). Since 2000, the local government has

delivered intermittent supplies of ecological water from Boston Lake to the lower reaches. During April to September 2012 and April to September 2013, 650 and 504 million m$^3$ of water were transferred to the lower reaches of the Tarim River, which had a positive effect in raising the groundwater level and restoring riparian vegetation (Li et al. 2005; Chen et al. 2015). However, the groundwater levels in the middle and lower reaches were still much lower than in the upper reaches. In





general, both surface and ground water availability were poor in the downstream oases. Better agricultural water allocation

strategies should be considered for integrated water resource management in the whole catchment.

## 3.2 Natural flooding through ecological gates

Due to low precipitation levels and water heads, natural flooding is crucial for the riparian forest along the river oases. There are more than 100 flood gates along the river. The designed flow rate varies from several to hundreds of cubic meters. In addition, local farmers also occasionally open-up small breaches on the river bank to flood their fields and trees. It is

therefore difficult to determine how much ecological water passes through all of these gates. We monitored several typical large gates in SC1, SC2, and SC3 (Fig. 6), then simulated the flooding rates by considering their capacities and thresholds.

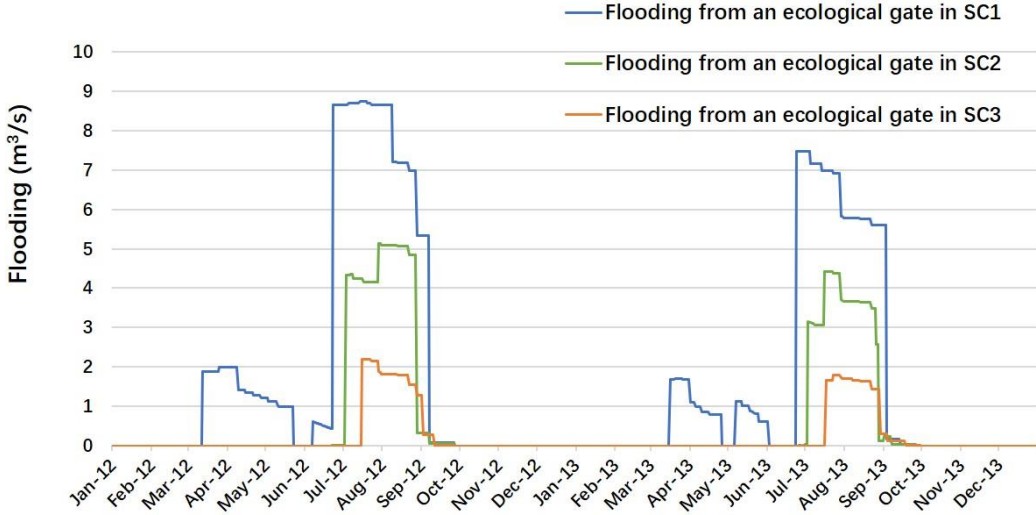

**Figure 6: Flooding through ecological gates in SC1, SC2, and SC3. Flooding mainly occurs in summer. In the downstream part of SC4, flooding through ecological gates is negligible.**

There was a clear decreasing trend in the flooding rate from the upstream to downstream ecological flood gates. The level of ecological flooding in SC3 was about 1/7 of that in SC1, which clearly indicated the shortage of ecological water in the middle reaches. In SC4 (lower reaches), due to the very low water volume and river interception, flooding from ecological flood gates was not included in the model. Spring flooding only occurred in SC1, and flooding in SC2 and SC3 only occurred in summer and early autumn. This is a critical condition for riparian forest in the downstream area considering the

groundwater levels. For example, for P. euphratica, adult trees will not be largely influenced as long as their roots can reach the water head, but young trees may not survive without ecological flooding. Therefore, the regeneration of Populus trees is threatened by the severe water scarcity problems in the middle and downstream oases. For grassland and scrub vegetation,





the living conditions are even more challenging because their roots are usually shorter than those of Populus euphratica. In SC4, plants mainly grow close to the river banks, and flooding from the ecological flood gates was not considered in the model.

### 3.3 Irrigation water consumption and water deficit

Irrigation water demand was calculated in the model by considering climate conditions, crop growing factors, and the cultivated areas. The actual irrigation water consumption was dependent on the water availability for different water users. Whenever both surface and groundwater were unable to supply the irrigation water demand, a certain water deficit would occur. Land management decisions regarding the reclamation or abandonment of farmland are mainly determined by water and land productivity, which are further constrained by the water deficit along the river oases. The simulation results for total irrigation water consumption and the water deficit of each sub-catchment are shown in Fig. 7.

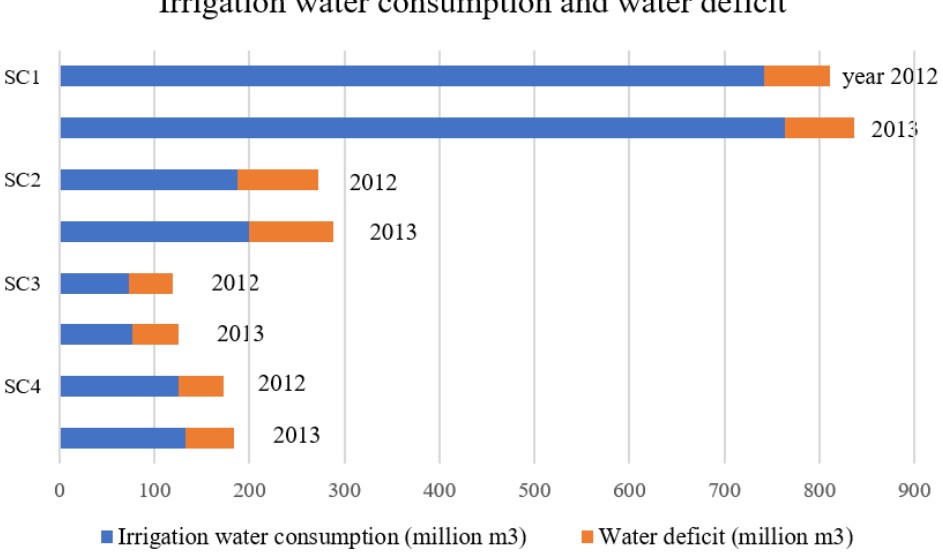

**Figure 7: Irrigation water consumption and water deficit in each sub-catchment in 2012 and 2013.**

Most of the farms in the study area were located in SC1, and this sub-catchment therefore consumed the largest share of irrigation water. All sub-catchments had a certain amount of water deficit, varying from 45 to 90 million $m^3$, but the ratios of water deficit to irrigation water consumption were much higher in SC2, SC3, and SC4 than in SC1. This indicated that for a certain unit of farmland, water deficits were larger in the middle and lower reaches than in the upper reaches. A peculiar phenomenon was also found, in which as the water consumption increased from 2012 to 2013, the water deficit was not reduced. After further investigation, this was found to be due to the increase in farmland area. In the study area, crop rotation and leaving some land fallow are very common practices due to water scarcity. The area of farmland to cultivate each year is

largely dependent on the upstream river discharge and government water policies. In 2013, an additional 49.15 km$^2$ of farmland area was cultivated compared to 2012, which led to an increase in the total irrigation water demand.

### 3.4 Land use changes

Under the future climate scenario CCLM RCP 2.6 (Duethmann et al. 2016), water movements and vegetation growth were simulated and land use changes were predicted until 2050. In this scenario (Fig. 8), our research focused on the middle reaches where most of the natural vegetation was located. Based on the logic and rules of the model, if a number of vegetation cells vanished in future years, it would mean that the groundwater level was too low for plant growth in the corresponding areas. Furthermore, it would indicate that ecological flooding was not sufficient and groundwater recharge

was too low to support natural vegetation growth in these areas. The farmland area would not change naturally in the DSS, because there is a basic rule in the model that if surface water is not sufficient for the farmlands, groundwater will be pumped to guarantee the irrigation water supply.

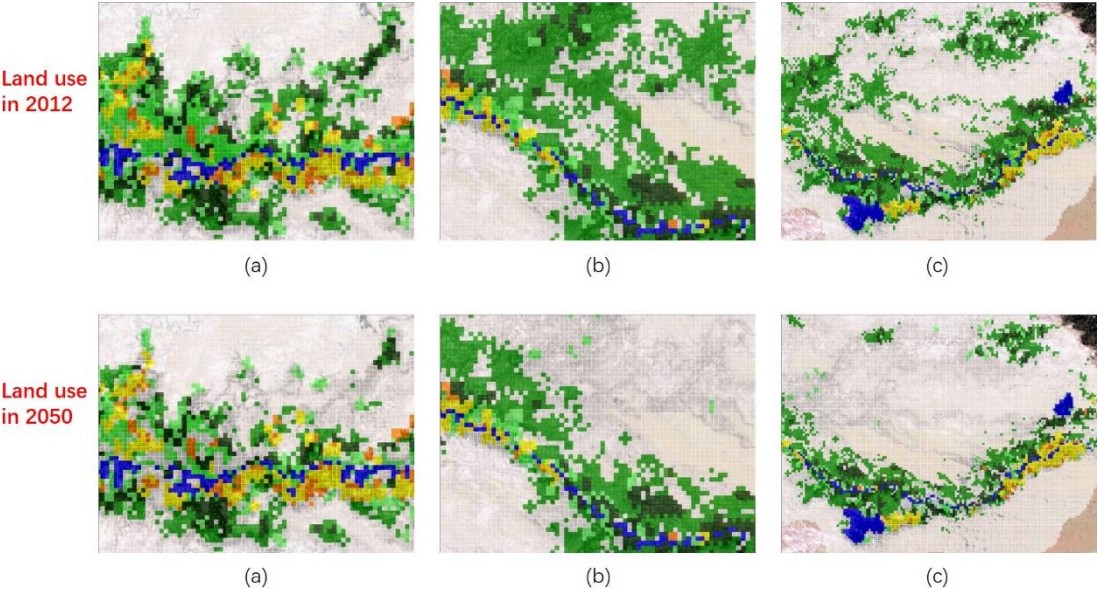

**Figure 8: Simulated land use changes from 2012 to 2050.**

Based on the simulation results, land degradation was quite obvious along the river oases, especially in the middle and lower reaches. From the comparison shown in Fig. 8, it is noticeable that many "green areas" had vanished in 2050, with this phenomenon occurring more often in locations far from the river course. By the end of simulation period, 25.9% of the area of riparian vegetation had degraded. The DSS suggested that this degeneration of natural vegetation was caused by the lack of ecological flooding and low groundwater table. Fallow land and crop rotation have been applied for many years in the

study area, but the extra water resources in wet years would be better used to improve the decaying ecosystem in the downstream oases rather than in new cultivated farmlands. The simulation results clearly indicated ongoing land degradation





in the middle and lower reaches until 2050. To maintain the newly planted trees and grassland in the oases, additional water sources must be guaranteed in the next few years.

### 3.5 Ecosystem services

There are many possible scenarios in which the potential ecological water supply could be increased and the riparian ecosystem maintained (as indicated by ESS). In our model, the ESS were simulated under three different management scenarios (Fig. 9): (1) everything kept as usual (reference scenario); (2) the cotton area reduced by 20%; and (3) the extent of drip irrigation increasing (from 50% to 90%), while ecological flooding increased by 50%. The ESS changes were mainly dependent on management alternatives, water availability, and climatic variation during the simulation period. The resultant

impacts on socio-economic factors were closely associated with the well-being of human society and regional sustainable development. The ecosystem provisioning, regulating, and supporting services were weighted from 0 to 1, representing how many of our management goals could be fulfilled. These goals were generated from the expectations of local stakeholders and decision-makers, who expressed their suggestions on how to improve the ecosystem.

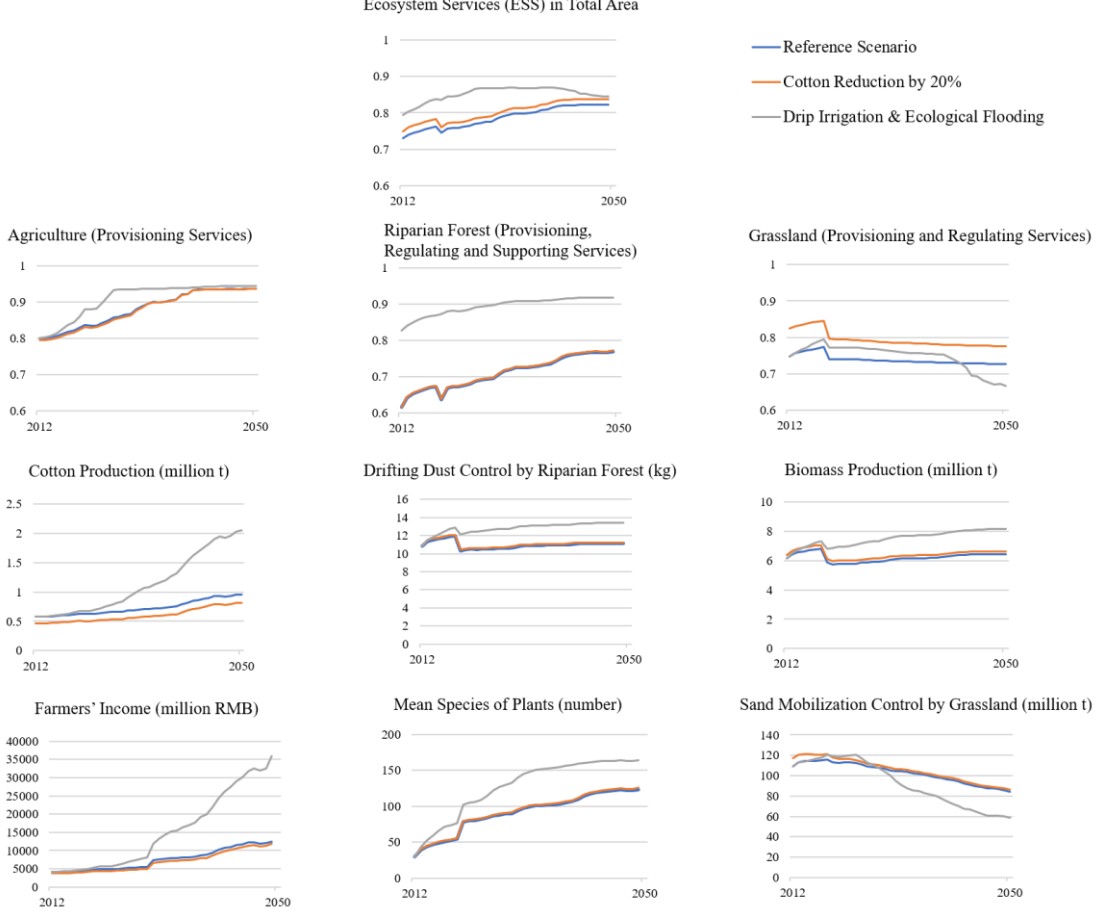



**Figure 9: Simulation results of ecosystem services, including detailed values for the classes of agriculture, riparian forest, and grassland.**

The results for the reference scenario indicated increasing trends in the total ESS values for agriculture, riparian forest, and grassland, with increases of 16.9%, 24.7%, and 12.8%, respectively from 2012 to 2050. However, grassland provisioning and regulating services were expected to decrease. A number of grassland areas are either located far from the

ecological flood gates or the groundwater table is too deep for the plant roots to reach. This situation was improved in the cotton reduction scenario because 20% of the cotton area could easily be replaced by natural grassland. Cotton production and farming incomes would not be substantially reduced by a reduction in the cotton area, which suggests that water supply is a bottleneck in the crop yield in the river oases. This point was proven by the drip irrigation and ecological flooding scenario, which resulted in large amounts of irrigation water being saved and more ecological water being available. Drip

irrigation is likely to be very effective in increasing cotton production and farming incomes in future years. An increase in ecological flooding would also improve the provisioning, regulating, and supporting services supplied by riparian forest (e.g., drifting dust control and the mean number of plant species). However, in the period to 2050, there would be a significant decline in agricultural provisioning services, grassland provisioning and regulating services, and the total ESS value in the drip irrigation and ecological flooding scenario. The reason for this was the soil salinization effect. Drip irrigation would

cause the accumulation of salt in the soil profile, which is harmful for crop growth in the long term. Cotton has a strong tolerance to soil salinity, but many other crops and fruits cannot resist high levels of salinization, and their yields would be substantially reduced based on our model. Overall, in comparison with the reference scenario, the cotton reduction scenario would slightly increase the total ESS, and the drip irrigation scenario would substantially increase the ESS in the near future. However, there would then be a large decline by 2050 due to soil salinization.

**4 Conclusion and discussion**

China is on the frontline of combating desertification and ecosystem protection. In arid regions of China, planting trees is promoted by the central government and is undertaken by social communities and the general public. It is not clear of this is an appropriate and sustainable way to expand the vegetation area in the dryland ecosystems of China. In this study, we applied hydro-ecological models to assess the provision of water resources and predict ESS in the largest inland river basin

in China. The first conclusion of our study was that the current groundwater and surface water resources are not sufficient to support the sustainability of ecosystems in their current state, and therefore the further expansion of natural vegetation is not possible. The simulation results indicated that riparian forests will remain prone to deterioration if no significant changes are made to guarantee an ecological water supply along the river oases.

The Tarim River has the typical seasonal characteristics of a dryland river. Summer floods dominate the river runoff each

year by supplying more than 3/4 of the total water volume. Surface runoff is quite low in spring and winter. Most of the water resources are consumed in the upper and middle reaches. The discharge in the downstream is less than 1/7 of that in





the upstream. A higher percentage of the total water is consumed in the upper and middle reaches in dry years, which can be very harmful for the ecosystem in the lower reaches. On the other hand, groundwater levels decline substantially from summer to winter, and from the upper to lower reaches. In SC4, the groundwater level can be lower than 10 m in winter.

Overall, water stress sets variable and challenging limits on agricultural development and ecosystem rehabilitation in this arid region.

Ecological flooding displayed a decreasing trend from upstream to downstream oases. Without additional ecological water, it is very difficult for riparian trees to regenerate in the middle and lower reaches. Irrigation water consumption in SC1 was much higher than in the other three downstream sub-catchments combined. In recent years, the Chinese government and

social communities have made huge efforts to plant trees and undertake ecological restoration, such as "transforming cropland into grassland" and "returning cultivated land to forest" in the Tarim River Basin. However, the current ecological water resources (including surface ecological water and groundwater) are not sufficient to maintain the riparian green belt in future years. Based on the simulated land use changes, more than 1/4 of the total vegetation area will be degraded by 2050 without the provision of additional ecological water resources.

The ESS scenarios evaluated in the DSS could lead to the provision of sustainable socio-economic and ecological benefits. The DSS enabled the interpretation of a complex range of environmental factors and the results were used to evaluate the current situation and propose creative solutions. The reference scenario indicated a 12.8% increase in the total ESS value, but a decrease in grassland provisioning and regulating services was also noticeable. The scenario with a reduced cotton area resulted in steadily increasing ESS values. This scenario was expected to have little effect on farming incomes, due to the

increased sales price of crops in future years. The drip irrigation and ecological flooding scenario has the best performance in terms of increasing the total ESS value in the near future. Cotton production, farming incomes, and riparian forest provisioning, regulating, and supporting services were all significantly improved in this scenario. In the far future (i.e., 2050), the soil salinization problem should be seriously considered to avoid the deterioration of ESS in the oases.

*Data availability.* Data available from SuMaRiO project website: https://www.sumario.de/. The DSS is also free available on request.

*Competing interests.* The author declares that there is no conflict of interest.

*Author contribution.* MD, XC, AB and RY initiated the research project. YY, PH and MH established the hydro-ecological models. YY wrote the manuscript. HZ and RY performed revisions. JL, FZ and RY provided ideas and data in the research area. LS and YG contributed preparation of data sets and modeling assistance. All the co-authors provided discussion and insights into the research.



*Acknowledgements.* We are very grateful to the Sino-German collaboration research project SuMaRiO (Sustainable Management of River Oasis along the Tarim River) funded by the German Federal Ministry of Education and Research (BMBF).

*Financial support.* This study has been supported by the "High level talent introduction project of the Xinjiang Autonomous
Region" (Y942171), the Chinese Academy of Sciences (CAS) "Light of West China" Program (2018-XBQNXZ-B-017) and the "100 Talents Program of the CAS" (Y931201).

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
