# Peer review of "Assessing ecosystem services under water stress in the largest inland river basin in China based on hydro-ecological modeling"

_Hydrology and Earth System Sciences, 2020_

## Referee Comment (RC1) · Anonymous Referee #1 · 28 Apr 2020

This paper presents a case study of water resources management and ecosystem service protection for a major basin in western China. While the case study is of potential interest, the methods used are standard, and there seems to be no methodological innovation or research interest. In addition, the paper is a very long way from the standard required for presentation in an international journal such as HESS. A few examples are presented below. There are very basic problems with the presentation of material. For example, the catchment is not defined on a location map, and the area modelled seems to be a particular reach of river, but this is not discussed. There is no discussion of the relevant climatological or hydrological processes, and any associated modelling challenges. The sub-catchments modelled are not defined, nor is there

any discussion of the selection of the boundaries for the modelling, and the boundary conditions. Standard models are used, but there is no discussion of model parameterization, model calibration or model uncertainty, and there is no discussion of the data available to support the modelling. Where simulations are presented in Fig 5, the units are not defined. The models overlap in scope, but there is no recognition of associated problems and no discussion of how potential inconsistencies are handled. Where the ecosystem services DSS is presented, methods are not defined. e.g.153-155 'Tree species were determined by the fuzzy logic between groundwater level and the flooding of natural vegetation. Apocynum and reed production were influenced by groundwater level, groundwater salinity, and grazing area.'

Certain aspects of the water resources management are unclear. For example, 'ecological gates' are mentioned, without any clear explanation of the physical situation and any associated control rules. There is much vague writing – apart from the lack of definition of modelling methods, terms are used such as line 83 'a huge amount of investment'. In addition, the discussion of the regulatory context is presented as quite partial and subjective, rather than objective –e.g. 'Due to government determination and the aspiration of the people, a new era of ecosystem protection has been predicted to emerge in China. . . . . .'

One minor point of detail - Ref to Mcdonnell should be McDonnell
* * *

---

## Short Comment (SC1) · 29 Apr 2020

Dear referee, thank you very much for your valuable comments. Before thoroughly consideration on all the comments point-by-point, I would like to mention that the hydrological modeling processes, including model parameterization, calibration, boundary conditions and modeling challenges have already been discussed in our previous publications:

Yu, Y., Disse, M., Yu, R.D., et al. (2015). Large-Scale Hydrological Modeling and Decision-Making for Agricultural Water Consumption and Allocation in the Main Stem Tarim River, China. Water., 7, 2821-2839. Yu, Y., Yu, R., Chen, X., et al. (2017). Agricultural water allocation strategies along the oasis of Tarim River in Northwest China. Agricultural Water Management., 187, 24-36.

In addition, other aspects of water resource management can also be supported by authors from our research group:

Cyffka, B., Rumbaur, C., Disse, M., et al. (2013). Sustainable management of river oases along the Tarim River (P.R. China) and the ecosystem services approach. Geography, Environment, Sustainability., 6, 77-90. Duethmann, D., Menz, C., Jiang, T., et al. (2016). Projections for headwater catchments of the Tarim River reveal glacier retreat and decreasing surface water availability but uncertainties are large. Environmental Research Letters., 11, 054024. Patrick, K., Markus, D., Halik, ümüt. (2015). Effects of Land Use and Climate Change on Groundwater and Ecosystems at the Middle Reaches of the Tarim River Using the MIKE SHE Integrated Hydrological Model. Water., 7, 3040-3056.

This paper is a summarizing paper for the 5-year Sino-German collaboration research project SuMaRiO (Sustainable Management of River Oasis along the Tarim River). We try to present our important research outcomes which would give warnings to decision-makers and benefit on regional sustainability. I agree some parts of the article are too vague and unclear. Thank you very much for the comments. We shall carefully think about the problems and revisions in the next steps.

---

## Referee Comment (RC2) · Anonymous Referee #2 · 11 May 2020

The submission entitled by 'Assessing ecosystem services under water stress in the largest inland river basin in China based on hydro-ecological modeling' is well written with clear objectives and convinced results. Current water cycle and ecosystem protection measures were simulated, and future land use change scenarios were proposed accordingly. China is on the frontline of ecosystem protection and afforestation, but according to the simulation results, the available water resources cannot support more vegetation in its largest inland river basin. Without an additional water supply, 25.9% of the existing area of natural vegetation will be degraded by 2050. After reading the manuscript, I would like to give a few comments to improve the quality of the submission.

1. Ecosystem services should be shorted as ESs instead of ESS. 2. Due to model complexity and computational time, it is very difficult for a single model to consider both hydrological processes and ESS changes. Two hydrological models (MIKE HYDRO and MODFLOW) were employed simulate hydro-ecological processes and assess ESS changes, but the theory of ESs assessment is still not clear in current version, please introduce more details about the methodology to assess ESs and include necessary references. 3. How to calculated carbon storage, wind erosion control, dust control services, please introduce more details and include necessary references. 4. Did you do the comparison between ESs estimated by your model with previous studies? 5. Did you validate the ESs estimation results? 6. Both the MIKE HYDRO and MODFLOW models were fully calibrated and validated to precisely simulate the water cycle, but the ESs estimation results should also be validated. 7. Please declare your main objectives of this study in a clear and concise manner in Introduction Section. 8. The figures made by Excel should be replotted by other software, because they are ugly and no Y axis was clearly labeled in current version. 9. Conclusions and discussion should be written in two sections, and the current discussion is still shallow, please add more in depth discussion. 10. Conclusion should be declared in a concise and clear manner. 11. I do not think the ESs estimated by your model are reliable without validations and comparisons with previous results. 12. Explain more about the resources of each variables in Fig. 9, how did you get the outcome and whether they are convinced. 13. It is a big challenge to include all the things in one or two models, so how to combine ESs and hydrological process is still a big question that should be replied in your study, and more work is needed. 14. Section three should be results and discussions.

---

## Author Comment (AC1) · 4 Jul 2020

Authors' response to comments by Reviewer #1:

We have carefully read and thought about the comments. Firstly, we thank the reviewer's comments. For the enumerated problems and suggestions by the reviewer, our responses are as follows:

Comment: This paper presents a case study of water resources management and ecosystem service protection for a major basin in western China. While the case study is of potential interest, the methods used are standard, and there seems to be no methodological innovation or research interest.

Reply: The afforestation of China has drawn worldwide attention in recent years. But now a small fraction of researchers begin to question water availability under such large-scale tree-planting actions, especially in arid regions (e.g. Loess Plateau, Tarim Basin). Therefore, we used some standard methods to verify our first important result: current water availability is not able to support afforestation and ecosystem protection in a sustainable way in the largest inland river basin in China. Such results give warnings to local decision-making and policies. We think our results shall be interesting to certain governments, public and scientific communities. Another innovative part of our study is to combine hydrological models with ESs outputs. We used fuzzy logic, equations and expert knowledge, and complied them with C++ programs. We will be happy to add more details in the manuscript in this regard.

Comment: In addition, the paper is a very long way from the standard required for presentation in an international journal such as HESS. A few examples are presented below. There are very basic problems with the presentation of material. For example, the catchment is not defined on a location map, and the area modelled seems to be a particular reach of river, but this is not discussed.

Reply: the catchment is defined on the MIKE HYDRO map view. Because it is a relatively "simple" river catchment with no tributaries, we are not sure if another location map is needed. But more discussions on the basic information about the catchment would be added in the revised manuscript.

Comment: There is no discussion of the relevant climatological or hydrological processes, and any associated modelling challenges.

Reply: thank you very much for the suggestion. Some modeling challenges were already discussed by our previous publications. Besides, it is a big challenge to combine hydrological models with ESs outputs. We have put a lot of efforts on the logics and programming. We also think it is better to add more details in this regard.

Comment: The sub-catchments modelled are not defined, nor is there any discussion of the selection of the boundaries for the modelling, and the boundary conditions. Standard models are used, but there is no discussion of model parameterization, model calibration or model uncertainty, and there is no discussion of the data available to support the modelling.

Reply: boundary conditions, parameterization, model calibration, uncertainty and data availability were already included in our previous publications. But we will think about add more necessary information in this manuscript. Moreover, we think an additional validation process of our ESs model is very necessary. We prepare to add this validation process with our collected data to make our results more reliable.

Comment: Where simulations are presented in Fig 5, the units are not defined.

Reply: this is a mistake in the manuscript. We have revised it. Thanks for the comment.

Comment: The models overlap in scope, but there is no recognition of associated problems and no discussion of how potential inconsistencies are handled.

Reply: we agree the overlapping problems are big challenges and how to handle the potential inconsistencies is very necessary to be added in the manuscript. Actually, we have defined dozens of certain one-way interface between parallel models, and fetch outcomes from more specialized models (e.g. flooding from the MIKE HYDRO and land use change from the DSS). Indeed, more discussions are needed in the manuscript.

Comment: Where the ecosystem services DSS is presented, methods are not defined. e.g.153-155 'Tree species were determined by the fuzzy logic between groundwater level and the flooding of natural vegetation. Apocynum and reed production were influenced by groundwater level, groundwater salinity, and grazing area.'

Reply: the methods of ESS calculations were defined by fuzzy logic and equations. The standard methods were improved by our expert knowledges. More details on the methods would be added in the revised manuscript.

Comment: Certain aspects of the water resources management are unclear. For example, 'ecological gates' are mentioned, without any clear explanation of the physical situation and any associated control rules. There is much vague writing – apart from the lack of definition of modelling methods, terms are used such as line 83 'a huge amount of investment'. In addition, the discussion of the regulatory context is presented as quite partial and subjective, rather than objective –e.g. 'Due to government determination and the aspiration of the people, a new era of ecosystem protection has been predicted to emerge in China…...'

Reply: indeed, all the listed problems are necessary to be revised. Explanation of physical situation and control rules on ecological gates have been added in the manuscript. 'a huge amount of investment' has been changed into 'investment of over 4 trillion RMB'. 'government determination' has been changed into 'government policies', and 'the aspiration of the people' has been changed into 'public participation'. Besides, we have found another three parts with vague writing and subjective problems. We have revised them in the manuscript. Thank you very much for the comments and suggestions.

---

## Author Comment (AC2) · 4 Jul 2020

We thank the reviewer for the valuable comments and suggestions to improve our manuscript. Point by-point responses to the comments of are provided in the following text.

Comment: The submission entitled by 'Assessing ecosystem services under water stress in the largest inland river basin in China based on hydro-ecological modeling' is well written with clear objectives and convinced results. Current water cycle and ecosystem protection measures were simulated, and future land use change scenarios were proposed accordingly. China is on the frontline of ecosystem protection and afforestation, but according to the simulation results, the available water resources cannot support more vegetation in its largest inland river basin. Without an additional water supply, 25.9% of the existing area of natural vegetation will be degraded by 2050. After reading the manuscript, I would like to give a few comments to improve the quality of the submission.

Reply: we thank the reviewer for the positive comments on our paper. The afforestation of China has drawn worldwide attention in recent years. But now a small fraction of researchers begin to question water availability under such large-scale tree-planting actions, especially in arid regions (e.g. Loess Plateau, Tarim Basin). Our Sino-German cooperative research studied water conditions in the largest inland river basin in China. Our results clearly indicate the current unsustainable water-ecosystem nexus. Such results give warnings to local decision-making and policies. We think our results shall be interesting to certain governments, public and scientific communities. Thanks the reviewer for the kind words.

Comment: 1. Ecosystem services should be shorted as ESs instead of ESS.

Reply: indeed, ESs is the standard writing. We have revised it in the manuscript.

Comment: 2. Due to model complexity and computational time, it is very difficult for a single model to consider both hydrological processes and ESS changes. Two hydrological models (MIKE HYDRO and MODFLOW) were employed simulate hydro-ecological processes and assess ESS changes, but the theory of ESs assessment is still not clear in current version, please introduce more details about the methodology to assess ESs and include necessary references.

Reply: we agree more details about the theory of ESs assessment will improve the manuscript. The methodology mainly includes fuzzy logic, equations and expert knowledges, which are complied by C++ programs. Such methodologies will be added in the revised manuscript, along with necessary references.

Comment: 3. How to calculated carbon storage, wind erosion control, dust control services, please introduce more details and include necessary references.

Reply: carbon storage, wind erosion control and dust control services are calculated by fuzzy logics which are formed by our expert knowledge. We will introduce more details and add relevant references in the revised manuscript.

Comment: 4. Did you do the comparison between ESs estimated by your model with previous studies?

Reply: not yet. But this is a very good point and valuable suggestion. We will make some comparison with previous studies.

Comment: 5. Did you validate the ESs estimation results?

Reply: in the last 2 weeks, we have done a preliminary validation on the ESs estimation results (on several indicators) based on collected data. The preliminary results show good agreements. We prepare to conduct a thorough validation on all the indicators.

Comment: 6. Both the MIKE HYDRO and MODFLOW models were fully calibrated and validated to precisely simulate the water cycle, but the ESs estimation results should also be validated.

Reply: we also agree that this is very important, to make our results more scientific and convincible. In the next step, we will validate the ESs results.

Comment: 7. Please declare your main objectives of this study in a clear and concise manner in Introduction Section.

Reply: the main objective of this study is to find our whether water availability is able to support afforestation and ecosystem protection in a sustainable way in the largest inland river basin in China. The assessments of ESs will help us evaluate current situations and achieve socio-economic and ecological benefits in a sustainable way. The main objectives will be added in the revised manuscript.

Comment: 8. The figures made by Excel should be replotted by other software, because they are ugly and no Y axis was clearly labeled in current version.

Reply: we agree to replot the Excel figures by other software (e.g. MATLAB), to make the figures clear and standard.

Comment: 9. Conclusions and discussion should be written in two sections, and the current discussion is still shallow, please add more in depth discussion.

Reply: we agree to separate conclusions and discussion into 2 sections. Indeed, it is very important to explain our results and add more discussion. We will include more in depth discussion in the revised manuscript.

Comment: 10. Conclusion should be declared in a concise and clear manner.

Reply: yes. As the conclusion and discussion sections would be separated, the conclusion will be revised in a concise and clear manner, to improve the manuscript.

Comment: 11. I do not think the ESs estimated by your model are reliable without validations and comparisons with previous results.

Reply: we agree. We prepare to make a thorough validation for the ESs results with collected data, and we will compare the results with previous studies in the revised manuscript.

Comment: 12. Explain more about the resources of each variables in Fig. 9, how did you get the outcome and whether they are convinced.

Reply: we will explain more about the variables in Fig.9. Cotton production, farmer's income, drifting dust control, mean species of plants, biomass production and sand mobilization control are

calculated by fuzzy logic and equations. The standard methods were improved by our expert knowledges. More details and references would be added in the revised manuscript.

Comment: 13. It is a big challenge to include all the things in one or two models, so how to combine ESs and hydrological process is still a big question that should be replied in your study, and more work is needed.

Reply: we agree. It is a big challenge to combine hydrological models with ESs outputs. This is an innovative part of our study. We used fuzzy logic, equations and expert knowledge, and complied them with C++ programs. We will add more details in the manuscript in this regard, and an ESs validation process will be performed.

Comment: 14. Section three should be results and discussions.

Reply: we also think it's better to make section 3 results and discussions. We are truly very grateful for the reviewer's comments that will help us largely improve the quality of our manuscript.